# A functional approach towards the design, development, and test of an affordable dynamic prosthetic foot

Mathieu Falbriard[1]*, Grégory Huot[2,3], Mathieu Janier[4], Rajasundar Chandran[5,6], Michael Rechsteiner[3], Véronique Michaud[6], Joël Cugnoni[4,7], John Botsis[4], Klaus Schönenberger[2], Kamiar Aminian[1]

1 Laboratory of Movement Analysis and Measurement (LMAM), EPFL, Lausanne, Switzerland, 2 EssentialTech Center, EPFL, Lausanne, Switzerland, 3 Physical Rehabilitation Programme, ICRC, Geneva, Switzerland, 4 Laboratory of Applied Mechanics and Reliability Analysis (LMAF), EPFL, Lausanne, Switzerland, 5 iPrint Institute, HEIA-FR, HES-SO University of Applied Sciences and Arts Western Switzerland, Fribourg, Switzerland, 6 Laboratory for Processing of Advanced Composites (LPAC), EPFL, Lausanne, Switzerland, 7 Institute of materials and mechanical design (COMATEC), School of Engineering and Management, University of Applied Sciences and Arts- West Switzerland (HES-SO), Yverdon-les-Bains, Switzerland

télThese authors contributed equally to this work.
* mathieu.falbriard@epfl.ch

**Data Availability Statement:** All relevant data are within the paper and its Supporting Information files.

## Abstract

Humanitarian actors involved in physical rehabilitation, such as the International Committee of the Red Cross (ICRC), usually provide their beneficiaries with lower-limb prostheses comprising Solid Ankle Cushion Heel (SACH) feet as these are considered appropriate (price, durability, low profile to fit a majority of patients, appearance) and reliable for all ambulation levels. However, individuals in low-resource settings having higher ambulation abilities would greatly benefit from dynamic prosthetic feet with improved biomechanics and energy storage and release. Some attempts tried to address this increasing need (e.g. Niagara Foot) but most products proposed by large manufacturers often remain unaffordable and unsuitable to the context of low-resource settings. The design requirements and a price target were defined in partnership with the ICRC according to their initial assessment and used as a starting point for the development process and related technological choices. Numerical simulation and modeling were used to work on the design and to determine the required materials properties (mechanical, chemical, wear), and a cost modeling tool was used to select suitable materials and relevant processing routes (price vs. performance). A prosthetic foot comprising an internal keel made of composite materials, a filling foam, and a cosmetic shell with a foot shape was developed. Manufacturing processes meeting the cost criteria were identified and prototype feet were produced accordingly. These were successfully tested using a compression testing system before gait analyses were performed in the laboratory with non-amputees wearing testing boots. After validation in laboratory conditions, the prototype foot was tested in the field (Vietnam) with 11 trans-tibial unilateral amputees, who showed an increased mobility compared with the SACH foot. The collaboration of different research fields led to the development of a prosthetic foot which met the technical requirements determined by the ICRC's specific needs in its field of operation. The materials

**Funding:** MF, GH, MJ, and RC were funded by the Swiss Commission for Technology and Innovation (CTI) through grant 25441.1 PFIW-IW "Advanced ICRC prosthetic foot."

**Competing interests:** The authors have declared that no competing interests exist.

and selected production processes led to a manufacturing cost of less than 100 USD per part.

## 1 Introduction

According to the World Health Organization (WHO), 80% of people with disabilities live in low- and middle-income countries (LMICs) [1]. Lower-limb amputees in LMICs, which number is expected to increase due to the increasing diabetes prevalence globally [2], face two challenges: the limited access to rehabilitation services [1] and the low availability of prosthetic components which are appropriate to the context of use. Particularly, prosthetic feet (further described as "feet" for ease of reading) with advanced biomechanical features (i.e., inversion/eversion, plantar/dorsal flexion, energy storage and release), also known as "dynamic feet", are usually made of expensive materials such as autoclaved carbon fiber prepregs that make them out of reach to a majority of people living in LMICs. Therefore, despite being needed for patients with higher mobility potentials, there is a lack of affordable dynamic feet suited for low-income countries [2]. Some commercial feet (e.g., Blatchford SuperSACH foot, Jaipur foot) try to address this need but have limited functionality and do not allow for activity level 3 (ambulation with variable cadence, ability to traverse most environmental barriers) which is expected from dynamic feet. Activity levels, also known as K-levels, range from 0 (no ambulation ability) to 4 (exceeds basic ambulation skills) in the US Medicare classification [3]. The Niagara Foot [4, 5] also targets the same audience and is suitable for level 3 users. Unfortunately, this foot only fits a limited range of patients due to its high profile (higher than a SACH foot, not suited to patients with longer stumps) and limited sizes available for both the keel and the cosmesis. Recent studies proposed innovative solutions [6–8], but the design process rarely included optimizing the foot production costs at each step of the conception, nor did they include a functional evaluation.

As a major actor in the rehabilitation of lower-limb amputees in conflict-affected areas and LMICs, the International Committee of the Red Cross (ICRC) aims to provide affordable prostheses to the highest number of beneficiaries. Hence, through its Physical Rehabilitation Programme (PRP), the ICRC developed the Polypropylene (PP) Technology, a well-adapted and reliable solution for the specific context of low-resource settings. Over the past decades, the ICRC has fitted lower-limb amputees with CR-SACH feet (CR Equipements SA, CH), a cost-effective, solid-ankle cushion-heel foot suited for mobility levels 1 and 2 (K1–K2) as it features limited biomechanical properties. According to the ICRC, no K3 feet on the market currently meet their needs in terms of price, profile height, durability, and appearance.

Therefore, this study aimed to design, manufacture, and test a dynamic foot according to the following specifications: (1) The total manufacturing cost should be maximum 100 USD. (2) The foot must be compatible with the PP Technology (i.e., can be bolted to the prosthetic leg through the foot arch). (3) The targeted performance should meet the Medicare level K3, which corresponds to WHO's International Classification of Functioning, Disability and Health code d4602 [9], and take into account users weight range. (4) The mechanical performance should comply with the standard ISO 10328:2016 and relevant P levels (the prototype foot targeted maximum 100kg users, corresponding to testing level P5) [10]. (5) The foot must comprise a fully-encapsulated keel in a real-looking non-removable cosmetic shell with a split toe to ensure durability (i.e., the cosmetic shell acts both as a protection against environmental and wear factors and to decrease the stigma of disability); with external dimensions, especially

the profile height, of the foot being similar to the ones of the CR-SACH to fit the maximum users possible (stump length).

## 2 Materials and methods

### 2.1 Project framework

The project consisted of two main phases (Fig 1): (1) understand the needs and set the objectives, and (2) design a foot according to the requirements set in (1). Each phase was divided into inter-related sub-tasks, which were, in some cases, repeated several times within the course of the study.

Based on ICRC's needs, an analysis of the value chain and the future foot's life cycle was carried out using EssentialTech's (EPFL) methodological approach [11]. This method clearly defined the objectives of the project and identified the resources required. Hence, a team of three EPFL laboratories was set up to face the challenges related to biomechanics and gait analysis (LMAM), mechanical design and simulation (LMAF), and materials science, composites processing and technical cost analysis (LPAC). During the project's course, this approach, proposed by EssentialTech, continually adjusted the research efforts to the needs of the beneficiaries, and other stakeholders, in close collaboration with the ICRC physical rehabilitation team.

To gain insight into the mechanical response of existing feet, we carried out a set of in-lab gait analysis tests and mechanical characterizations. Four feet were selected based on their Medicare classification (K-level) and their range of applications [3, 12]. These were: $P_{SACH}$ (CR-SACH, CR Equipements SA, CH) a solid ankle cushion heel foot, not dynamic, activity level K1-K2, low profile, polymer keel encapsulated in foam and widely used for humanitarian applications; $P_{K3\_C}$ (Model 2, Niagara, CA) a dynamic foot, activity level K2-K3, high profile, polymer keel in cosmesis and designed for humanitarian purposes; $P_{K3\_E}$ (1D35, Ottobock, DE) a dynamic foot, activity level K2-K3, low profile, short glass fiber-reinforced composite keel encapsulated in a foam; $P_{K4}$ (Vari-Flex, Össur, IS) a dynamic foot, activity level K3-K4, high profile, long carbon fiber-reinforced composite keel in a cosmesis. The results from this characterization process were used to set the mechanical requirements of the new foot.

After the objectives were defined during the first phase, the design of the prototype foot ($P_{PRO}$) was started. The design of $P_{PRO}$, based on Finite Element (FE) simulation, used the feedback from the in-lab biomechanical (i.e., gait analysis, load vs. pitch angle) and mechanical (i.e., reaction load & moment vs. deflection) evaluations to improve the model. These

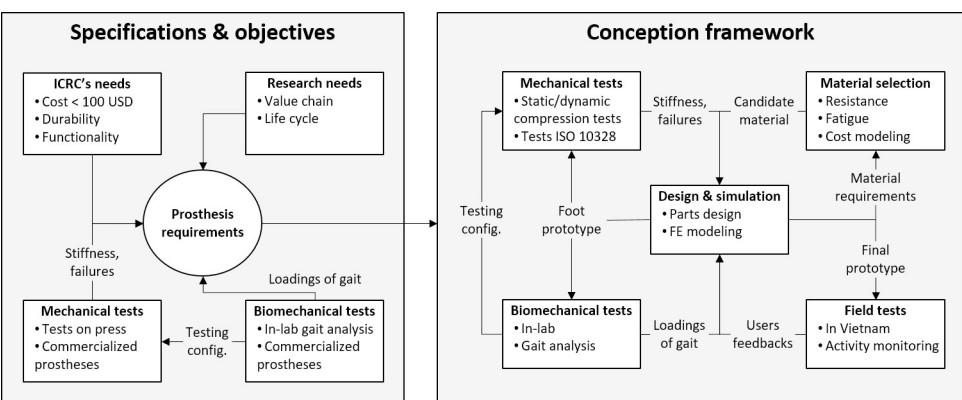

**Fig 1. The framework of the project.** This scheme shows the general structure of the project and the interactions between the different tasks. See the text for details.

simulations also influenced the materials selection process, which identified candidate materials based on their cost and mechanical properties.

Several design iterations were required to obtain a satisfying version of $P_{PRO}$ that would meet all requirements and objectives. This final version was tested in Vietnam with actual prosthetic users and provided valuable feedback for future improvements.

## 2.2 In-lab characterization of commercial prostheses

**2.2.1 Protocol.** In total, 13 non-amputee volunteers (age: 28 ± 8 years; weight: 74.6 ± 7.6 kg; size: 174.8 ± 6.1 cm) participated in the study. Each participant wore a pair of 2.4kg orthotic boots (Body Armor Walker 2, DARCO International, USA) (Fig 2), similar to those previously validated for gait analysis [13]. The participants performed a protocol composed of the following tasks: (1) alignment of the prosthetic system, (2) habituation period (minimum 15min), (3) sensor calibration, (4) level walking, (5) stairs climbing and (6) ramp climbing both frontally and laterally. These tasks were repeated for each foot (i.e., $P_{SACH}$, $P_{K3\_C}$, $P_{K3\_E}$, and $P_{K4}$) with the feet selected in a randomized order for each participant. A certified orthotist/prosthetist performed the alignment of the prosthetic system, and the order of the feet was randomized for each participant. Throughout the course of the project, three iterations of the biomechanical tests were performed (Fig 1): during the first iteration, the four commercial feet were involved, while in the second and third tests, different versions of $P_{PRO}$ were evaluated. The protocol has been approved by EPFL's ethical committee (HREC 001–2017) and was designed in agreement with the Declaration of Helsinki. Written informed consent was obtained from all the participants prior to the measurements.

**2.2.2 Measurement systems.** The kinematics and kinetics of the foot were recorded using several measurement systems. A ground-integrated force plate (Kistler, CH) measured the three-dimensional ground reaction forces (GRF) at 200 Hz during level walking. A stereo-photogrammetric motion tracking system with 11 cameras (Vicon, UK) recorded at 200 Hz the position of 20 reflective markers affixed on the prostheses. One inertial measurement unit (Physilog 4, GaitUp, CH) was placed on the dorsum of each foot and set to measure the

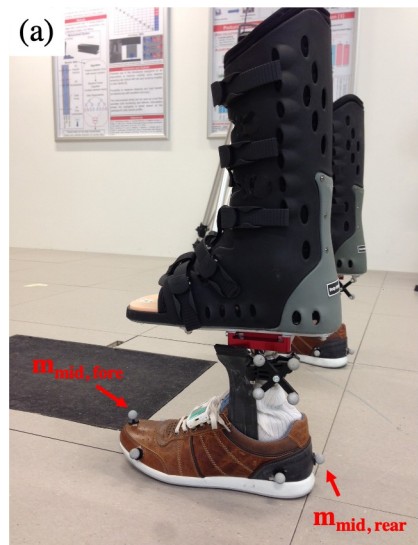 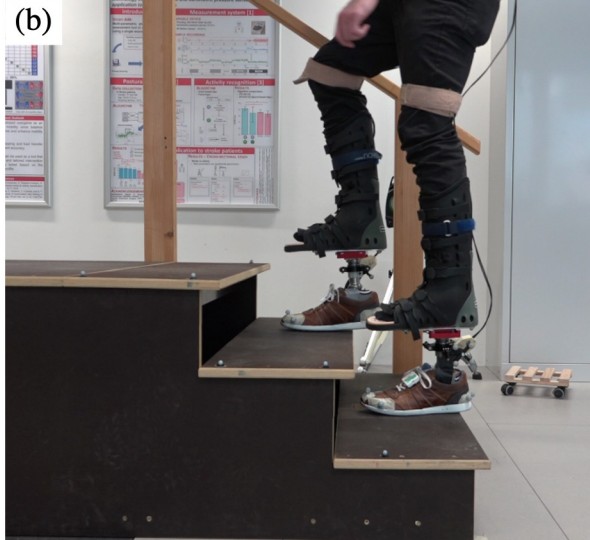

**Fig 2. Sensor configuration and protocol environment.** (a) the prosthetic system composed of the foot fixed under the orthotic boot. (b) The prosthetic system as worn by the participants during stair climbing.

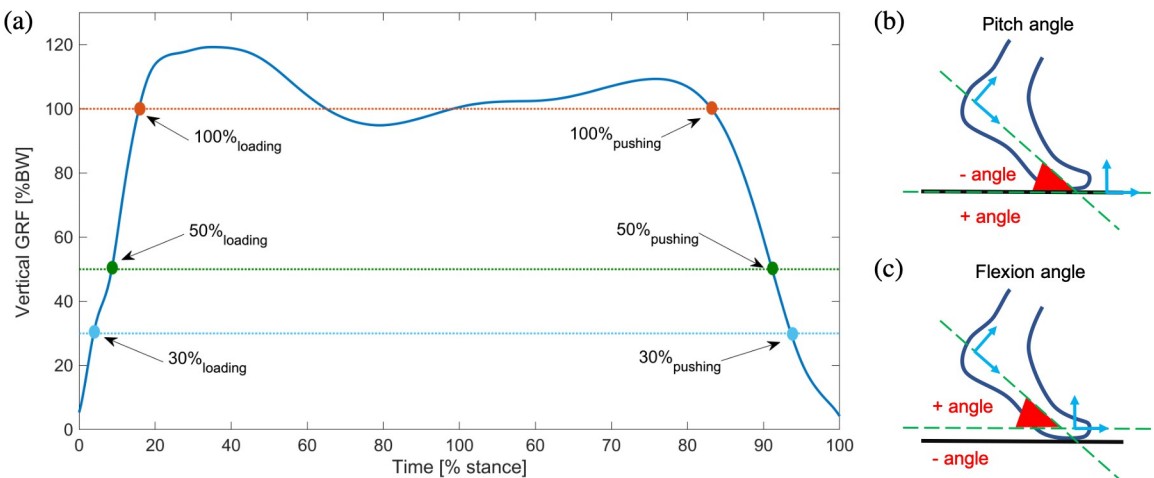

**Fig 3. Segmentation of the stance phase and angle definitions.** (a) Segmentation of the stance phase based on the normalized vertical ground reaction force. (b) Diagram of the pitch angle ($\alpha_{pitch}$) defined as the angle between the longitudinal axis of the rearfoot segment and the ground surface. (c) Diagram of the flexion angle ($\alpha_{flexion}$) defined as the angle between the vertical axis of the rear foot segment and the forefoot segment.

acceleration and angular velocity at 200 Hz (Fig 2). The data from these units were used to empirically confirm that the previously developed algorithms for sound leg gait analysis [14, 15] were functioning on the prosthetic foot and could be used for field tests. Pressure insoles (Pedar, Novel, DE) recorded the plantar pressure at the interface between the prosthetic foot and the sole of the shoe at 100 Hz. All measurement systems were electronically synchronized using a 5V trigger impulse.

**2.2.3 Gait analysis.** Standing still posture measurements were used to define the shank, fore and rear foot frames with the vertical axis perpendicular to the ground surface and the longitudinal axis parallel to the line connecting the $m_{mid,fore}$ and $m_{mid,rear}$ markers (Fig 2). We defined the pitch angle ($\alpha_{pitch}$) as the sagittal plane angle between the longitudinal axis of the rear-foot segment and the ground (Fig 3B). We also defined the flexion angle ($\alpha_{flexion}$) as the sagittal plane angle between the fore and rear foot frames (Fig 3C). The stance phases during gait were detected based on the vertical GRF (vGRF) obtained from the force plate [14].

For each foot, we computed the mean values of the vGRF, $\alpha_{pitch}$, and $\alpha_{flexion}$ for all the steps of each trial, and then calculated the mean ± STD over all subjects. However, we used the median, and the interquartile range (IQR) as inter-steps statistics (i.e., calculated over all the steps regardless of the subject) as histograms and the quantile-quantile plots (Q-Q plots) suggested a non-Gaussian distribution. To further characterize the feet, their behavior during the stance phase was assessed at 30%, 50%, and 100% of body weight (BW) during the loading and pushing phase, respectively (Fig 3A). For each foot, the values of the average angles ($\alpha_{pitch}$ and $\alpha_{flexion}$) were estimated at the three *Loading*, and the three *Pushing* times over all the steps and median ± IQR was then calculated over all subjects. Since we considered $P_{K4}$ as the foot with the highest performance, a pairwise distribution difference between $P_{K4}$ and the other feet was calculated using the non-parametric Kruskal-Wallis test.

## 2.3 Design, materials, and simulation

**2.3.1 Concept design.** We divided the design of the keel into three elements: a blade, an ankle, and a filling foam (Fig 4A; (iv), (iii), (ii)). We then encapsulated these elements into the CR-SACH cosmetic shell (Fig 4A; (i)). The foot is connected to the rest of the prosthetic leg

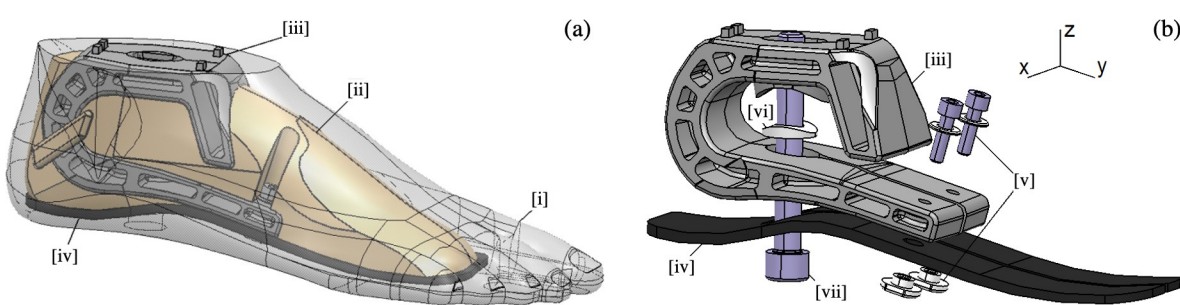

**Fig 4. Design of the prototype foot.** (a) Representation of the prototype foot composed of (i) a cosmetic shell, (ii) a filling foam, (iii) an ankle, and (iv) a blade. (b) PP Technology's bolting system to connect the foot to the rest of the prosthetic leg composed of (v) an M10 bolt and (vi) a specific washer, (vii) ankle, and blade fixation with M5 bolts and inserts.

with a removable M10 bolt (Fig 4B; (vii)) passing through a 20 mm hole at the foot arch. The design of the keel was based on three main criteria: cost, structural integrity (ISO 10328:2016), and the targeted performance (d4602 [9]) for an 80 ± 10kg patient using a 26 cm size foot. Also, the heel was designed 10 mm higher than the forefoot to fit the cosmetic shape. This design took into account the strength limits and stiffness of each part and was optimized to exhibit the best deflection vs. pitch angle response based on reference measurements obtained with commercial dynamic prostheses (see Results, Fig 8).

Since the ankle must allow high elastic deformation during the forefoot loading case, we opted for a "C" shape design to maximize the bending length. We added a security system to create a mechanical contact between the upper and lower part of the ankle to avoid an overload failure.

The blade was set to be mechanically constrained during the gait cycle; it works as a beam on the heel and the forefoot. The strength and the elastic behavior of the blade were optimized following several set targets: minimal use of carbon fiber reinforced material to limit cost (the target cost of this part was 35 USD), geometrical constraints due to the presence of the split toe, flexural fatigue resistance of 2 million cycles at 750 MPa stress conditions, and a flexural modulus in the range defined by the observation of commercial feet, to 100 GPa. Mainly, three parameters were explored: the general blade shape, thickness, and carbon fibers orientation in the layup (as well as the type of carbon fiber with good price/intermediate modulus and high strength). Both the C-shaped ankle and the blade feature a slit in the forefoot to allow eversion and inversion up to 15˚.

Furthermore, the foam had three functions: (1) filling the volume inside the foot to stabilize the outer cosmetic shell, (2) preventing the penetration of liquids or dirt while being light, and (3), when the foam reaches its compaction point, it also limits the deformation between the blade and the ankle. Hence, changing the foam density allowed to adjust the overall foot rigidity based on the user's target weight.

Finally, we further reduced the costs by decreasing the number of items to be produced; the parts were symmetric and could therefore, be encapsulated in either left or right cosmetic shells. To respond to a various range of patient characteristics, the parts have been designed in 5 different sizes 22, 24, 25, 26, and 28.

**2.3.2 Materials and process selection methodology.** Different materials were screened for the three elements of the keel, namely the blade, ankle, and foam parts (Fig 4), based on the mechanical property requirements and possible processing methods guided by the cost and design constraints. A manufacturing cost modeling was performed based on the activity-based technical cost model described by Wakeman et al. [16] and previously implemented in industrial composite manufacturing projects at EPFL [17, 18]. This methodology takes into

consideration all material costs, energy costs, labor costs, machine depreciations and various practical aspects get to a realistic cost assessment. Additional details are provided in the supplementary information S2 Dataset on the cost analysis and various materials tested in this research, scalability for different feet sizes and stiffness requirements were taken into account for the materials and processes selection, together with the targeted volume of production of the parts (in the range of 5000 parts per year per size). Specimens (100 mm long, 10 mm wide, and thickness varying from 4–4.2 mm) of the ankle and blade materials were prepared and tested for fatigue performance using a mechanical testing machine (ElectroForce 3400 from BOSE, USA). Three-point bending loading mode was used with a span to length ratio of 16:1 and at 10 Hz frequency for the ankle materials and 5 Hz for the ankle and blade materials. Compression tests at 2 Hz and 70% deformation were carried out on cylindrical filling foam specimens (13 mm diameter and 10 mm height) to screen suitable candidates for the foam part and to provide the constitutive laws for FE analysis. The cosmetic shell was made of thermoplastic polyurethane (TPU) based foam formulations similar to and compatible with the filling foam material.

**2.3.3 Mechanical simulation.**   Non-linear FE analysis was carried out, in which all parts were simulated with different material properties with their relevant constitutive laws. Materials properties were characterized using quasi-static FE models reproducing compression and three-point bending tests to adjust material parameters until the experimental and numerical results matched. The cosmetic shell material was simulated using a polynomial hyperelastic model, which fitted the uniaxial test data carried out in the laboratory. The foam part was simulated using Ogden's hyper foam potential [19] where the model coefficient were identified from experimental compression test data. Continuous fiber composites (C.F.C) used for the blade were simulated using properties of one of the promising candidate materials, Gurit SE70 carbon-epoxy composite [20]. Long Fiber Thermoplastic (LFT) materials used for the ankle were simulated using an isotropic elastic modulus of E = 10.5GPa.

The foot stiffness and strength characterizations were performed using a quasi-static FE model reproducing a compression test on the entire foot assembly to evaluate the vertical load-deflection characteristics and ensure the integrity of the foot for different loading cases including ultimate (1) and fatigue (2) loads required to comply to the standard ISO 10328:2016. The ankle, the foam, and the cosmetic shell were meshed using quadratic solid tetrahedron elements. The blade, represented by a composite shell with a variable thickness, was meshed using a quadratic shell element. The final FE characteristic element size (~5mm) was determined using a mesh convergence study on the load-displacement response. The FE assembly was first oriented to model the desired loading angle and then clamped through a reference point, which was coupled to the upper horizontal surface of the prosthesis interface. The vertical displacement of a rigid horizontal plate representing the ground was simulated as in the experimental setup. Frictionless hard contact was used to model the foot-ground interaction.

The stiffness of the prototype was optimized manually with incremental modification of different variables in the CAD & FEA model, to reach a similar level of deflection and reaction moment (top of the ankle part (Fig 4A)) compared to the selected reference commercial feet for different stance phases (pitch angles) and load levels corresponding to gait loads (as depicted in Fig 8C). The optimization variables were the key ankle geometry dimensions such as: the skins and the ribs thicknesses; the foam type, the composite layup, and the blade's thickness distribution. The design constraints were the manufacturing limitations, such as, available foam, composite material grades and injection molding requirements. The position of the injection nozzle and the flow cross section surface have been optimized with the manufacturing partner in a way to avoid pressure drops before the end of the injection (molding flow

simulation). Finally the design satisfied the material strength criteria for both fatigue (2M cycles) and static overloading.

## 2.4 Prototype manufacturing

The prototype was produced in a four steps process which involved: (1) injection molding of the ankle part, (2) over-molding of the filling foam onto the ankle within a mold, (3) production of the blade by autoclave processing, followed by CNC machining and assembly to the ankle by fasteners, and (4) cosmetic shell molding over the keel-blade-foam assembly for complete encapsulation. The ankle, foam, and cosmetic shell injection molding were performed with an injection molding machine (Arburg Golden edition 470C, Germany) within an industrial context, based on the process recommendations from the material suppliers and the materials characterization completed in the EPFL laboratories. The composite blade prototypes were manufactured at EPFL, reproducing industrial conditions.

## 2.5 Mechanical compression tests

Feet stiffness and hysteresis were quantified and evaluated for reproducibility. Tests were performed using an axial-torsion servo-hydraulic mechanical testing system (MTS 809, USA) equipped with a 10kN load-cell, at a constant load rate of 800N/min. The vertical displacement and forces data were registered at 0.5Hz. Five cycles of load (800 N) and unload (10 N) were set using a sinusoidal signal. The feet were neutrally aligned on the frontal plane and tilted at different pitch angles: -30˚, -20˚, -10˚, -5˚, 0˚, +5˚, +10˚, +15˚, +20˚, with a mechanical system (Fig 5, V). We considered a negative pitch angle for the forefoot loading case and a positive pitch angle during heel loading (Fig 3B). To ensure a purely vertical force recording, hence without shear forces at the plantar surface, we placed a low friction linear bearing guide (Fig 5, II) with a Teflon tape under the foot. We tested $P_{SACH}$, $P_{K3\_E}$, $P_{K4}$, and $P_{PRO}$ in such a setting and compared their stiffness. These recordings were then compared with the nominal vertical ground reaction forces (NvGRF) observed during the in-lab gait analysis (Fig 3A). Four strain gauges were placed with 2 Wheatstone bridges (Fig 5, IV) to measure My and Mx moments.

The second series of compression tests were carried out on the prototype to meet the existing standard ISO 10328:2016, with P5 loading cases (user weight up to 100kg). Ultimate strength tests were performed at -20˚ and +15˚ consecutively until 4480N, with a constant load rate of 200N/sec and a flat step of 30 sec at the load.

Cyclic fatigue tests were performed with a brand-new prototype on a 3400 electro-force BOSE fatigue machine (BOSE, USA). Two million load cycles were applied successively on the forefoot (-20˚ pitch angle) and heel (+15˚) with a loading range of [50N – 1330N] and a frequency of 2Hz. As required by the standard, we performed a final compression test at 2240N at both angles with a constant load rate of 200N/sec and a flat step of 30 sec at the maximum load to verify the remaining strength of the prototype foot after fatigue.

## 2.6 Field testing

**2.6.1 Protocol.** The field test aimed to assess the real-life usability and performance of the new foot with actual prosthetic users and gain insight for further improvements. We recruited 11 healthy transtibial amputees (weight: 72 ± 11 kg, size: 170 ± 2 cm, amputation side: 6L and 5R, foot length: 25.5 ± 0.5) without other disabilities. The measurements took place at the Vietnamese Training Centre for Orthopaedic Technologists (VIETCOT, Hanoi, Vietnam), where participants were fitted by two certified orthotist/prosthetist with new transtibial prostheses (comprising a CR-SACH foot) two weeks before the data collection. Two feet, $P_{SACH}$ and $P_{PRO}$, were tested in a randomized order. The real-life usability protocol included: (1) checking

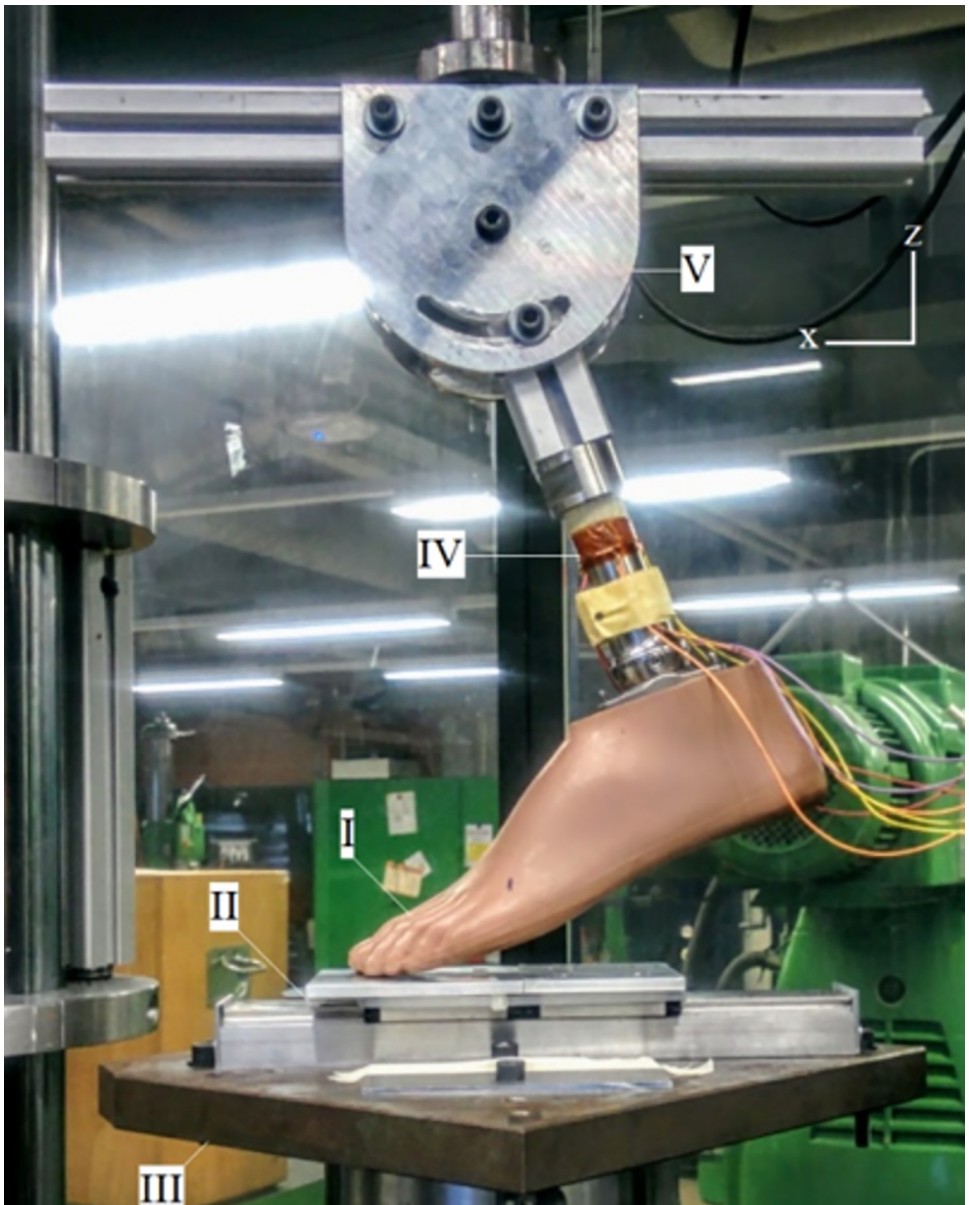

**Fig 5. Compression test setup.**

of the complete prosthesis, (2) habituation (~ 30min), (3) 6 Minute Walk Test (6MWT) [21] and (4) 48 hours of daily-locomotion monitoring using shoe-worn inertial sensors (Physilog5, GaitUp, CH). In addition, we used a foot assessment questionnaire (see S1 Appendix) adapted from the Prosthesis Evaluation Questionnaire (PEQ) [22] to evaluate comfort of the users based on (1) stability while standing, (2) stability while walking, (3) powerful push-off, (4) feeling of firm contact with the ground, (5) weight of the prosthetic foot, (6) comfort of the prosthetic foot and (7) overall score of the prosthesis. This protocol was accepted by the local ethical committee at the University of Labour and Social Affairs (ULSA, Vietnam) and written informed consent was obtained from all the participants prior to the measurements.

**2.6.2 Data analysis.** The detection of the locomotion periods and their classification for the daily measurement trials were completed using previously validated algorithms [23]. From

these algorithms, the number of walking bouts per day, and the mean duration and cadence of each bout was extracted as indicators of the daily physical activity of the patients [24–26]. The results were grouped by the foot type and the inter-trial mean, median, minimum, and maximum calculated for each of the aforementioned parameters.

The questionnaire (S1 Appendix) was composed of 14 explicit and 2 general questions. The mark of each question was reported relative to the length of the linear analog scale with a 100% score indicating a perfect evaluation. The inter-subject mean score of each question was calculated for each prosthesis, and results reported only for the questions where we observed a difference greater or equal to 2% (threshold empirically set based on the width of the pencil marks and the width of the scale).

## 3 Results

### 3.1 Biomechanical specification of PPRO versus commercial feet

In total, 840 steps (respectively: $P_{SACH}$: 205, $P_{K3\_C}$: 193, $P_{K3\_E}$: 195, $P_{K4}$: 192 and $P_{PRO}$: 55) of level walking have been analyzed with more than 10 steps recorded per participant and per foot. The average (min, max) number of steps per participant was 71 (51, 83) steps. This number varied between subjects as invalid steps (e.g., foot partially out of the force recording area, occlusion of the markers) were removed from the data set. The evaluation of the final foot prototype was performed during the third and final test, with only 7 of the 13 initial participants hence the lower number of steps recorded. The normalized mean ± STD of vGRF as a function of the pitch angle for each foot was compared and illustrated in Fig 6.

The inter-steps median and IQR of the six loading times of the stance phase and the corresponding $\alpha_{pitch}$ and $\alpha_{flexion}$ are shown in Table 1 with the p-values indicating a difference between each foot and $P_{K4}$ (highest profile).

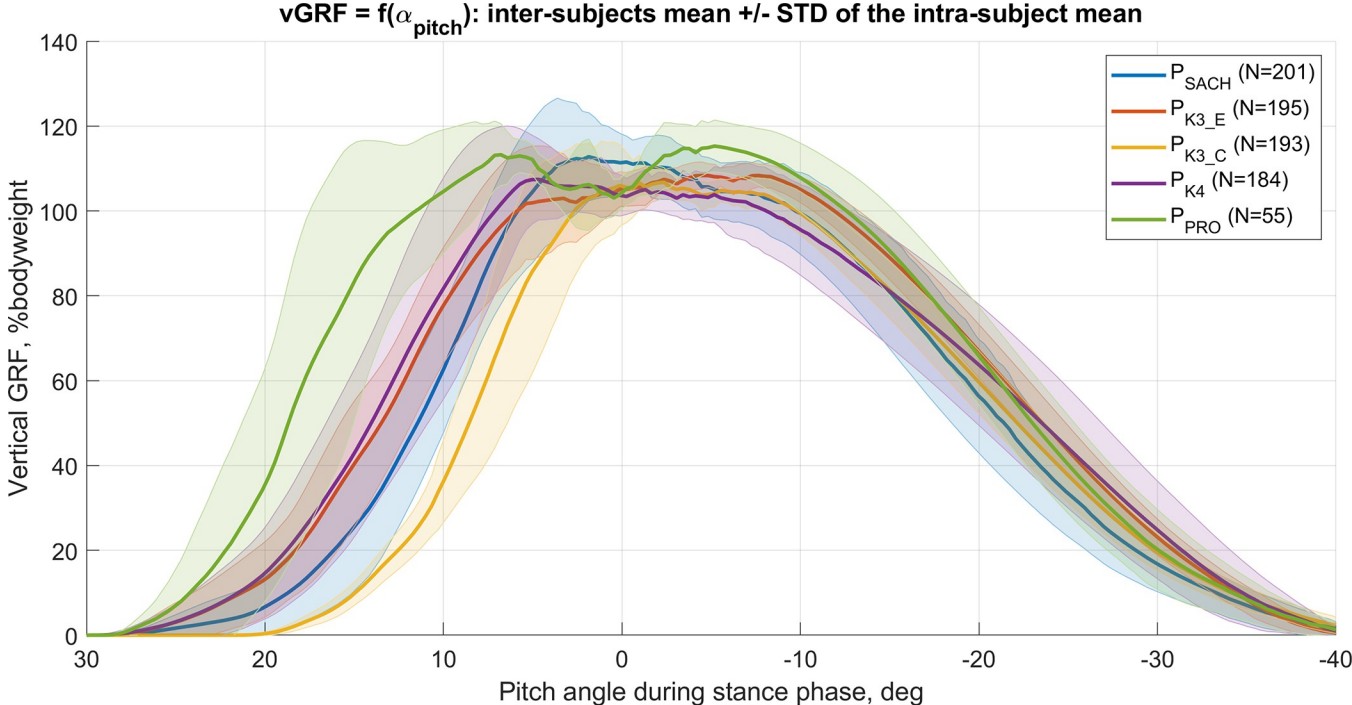

**Fig 6. Normalized vertical GRF as a function of $\alpha_{pitch}$ during within stance phase for five feet.** Each solid thick line corresponds to the mean values over all subjects, while the standard deviations are shown with shaded areas. The steps from 13 subjects were used to calculate $P_{SACH}$, $P_{K3\_C}$, $P_{K3\_E}$, and $P_{K4}$ statistics, while 5 subjects were used for $P_{PRO}$ (i.e., 5 among the 13 participants have tested $P_{PRO}$).

Table 1. Biomechanical specifications of four commercial feet and $P_{PRO}$[&].

| Parameters | vGRF [%BW] | $P_{SACH}$ (N = 205) | | $P_{K3\_E}$ (N = 195) | | $P_{K3\_C}$ (N = 193) | | $P_{PRO}$ (N = 55) | | $P_{K4}$ (N = 192) | |
|---|---|---|---|---|---|---|---|---|---|---|---|
| | | median | IQR | median | IQR | median | IQR | median | IQR | median | IQR |
| Pitch angle [°] | $30\%_{loading}$ | 15** | 7 | 16 | 8 | 11** | 6 | 21** | 6 | 17 | 8 |
| | $50\%_{loading}$ | 12** | 6 | 12 | 6 | 8** | 6 | 18** | 6 | 13 | 7 |
| | $100\%_{loading}$ | 8* | 7 | 7** | 8 | 4** | 6 | 14** | 8 | 8 | 8 |
| | $100\%_{pushing}$ | -10** | 6 | -12** | 4 | -10* | 5 | -12** | 6 | -9 | 9 |
| | $50\%_{pushing}$ | -20** | 6 | -23** | 4 | -22* | 5 | -22 | 4 | -22 | 9 |
| | $30\%_{pushing}$ | -24** | 6 | -28 | 4 | -26** | 4 | -26 | 5 | -27 | 8 |
| Flexion angle [°] | $30\%_{loading}$ | 4 | 6 | 3* | 5 | 4 | 6 | 0** | 3 | 4 | 6 |
| | $50\%_{loading}$ | 5 | 7 | 4* | 5 | 4 | 6 | 0** | 3 | 5 | 6 |
| | $100\%_{loading}$ | 5 | 7 | 3* | 4 | 5 | 6 | 0** | 3 | 4 | 5 |
| | $100\%_{pushing}$ | 8** | 6 | 7** | 4 | 4** | 5 | 8** | 6 | 4 | 7 |
| | $50\%_{pushing}$ | 13** | 5 | 11** | 3 | 9 | 5 | 10* | 5 | 8 | 5 |
| | $30\%_{pushing}$ | 13** | 5 | 10** | 3 | 9 | 4 | 8 | 4 | 8 | 4 |
| Time [%stance] | $30\%_{loading}$ | 3** | 2 | 3** | 2 | 4** | 1 | 2** | 1 | 3 | 2 |
| | $50\%_{loading}$ | 6 | 2 | 6 | 2 | 6** | 2 | 4** | 2 | 6 | 2 |
| | $100\%_{loading}$ | 10 | 5 | 11** | 8 | 9 | 5 | 7** | 5 | 9 | 3 |
| | $100\%_{pushing}$ | 84** | 3 | 85 | 2 | 84** | 2 | 87** | 2 | 85 | 4 |
| | $50\%_{pushing}$ | 91** | 1 | 92** | 1 | 91** | 1 | 92** | 1 | 93 | 1 |
| | $30\%_{pushing}$ | 93** | 1 | 95** | 1 | 93** | 1 | 95** | 1 | 95 | 1 |

[&]In this table, N is the number of steps used for the statistical analysis, and the non-parametric Kruskal-Wallis test p-value report significant difference with $P_{K4}$

* if $p < 0.05$ and

** if $p < 0.01$.

## 3.2 Prototype design, materials and process selection

**3.2.1 Prototype design.** Fig 7 presents the final design of a prototype foot, as well as the assembly showing the inner elements before the final over-injection step. After several iterations, the keel was successfully encapsulated in the cosmetic shell. The C-shaped front of the ankle ensures the integrity of the part in case of overload. This area has a gap of approximately 10mm, which closes when a high load is applied, limiting any structural failure in the ankle part in case of overloading.

**3.2.2 Materials and process selection.** After the cost modeling of each material and process selection step, the material choice, cost index, and preferred processes that satisfied the

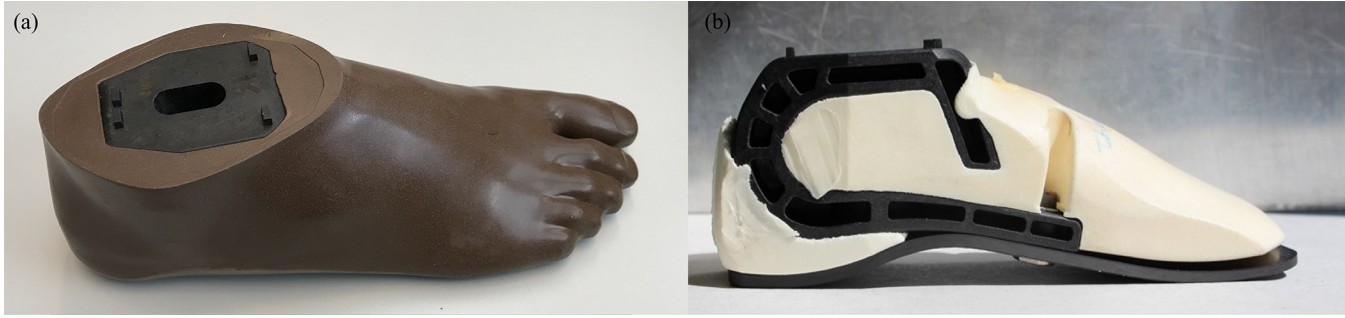

**Fig 7. The final aspect of the foot prototype.** (a) A prototype foot over-injected with a cosmetic shell. (b) Assembly of the keel, blade, and foam before the final over-injection.

**Table 2. Choice of materials and process satisfying the cost requirements[&].**

| Part | Requirements | Material choice | C.I., % of Target[*] | Preferred Processes |
|------|-------------|-----------------|---------------------|---------------------|
| Blade | Tensile Modulus: 100–130 GPa, fatigue resistance in bending up to 750 MPa to survive up to 2 million cycles | Continuous fiber composites: Primarily based on carbon epoxy prepregs | 35 | Composite lamination: Oven curing / Press molding |
| Ankle | Light, modular, cost-effective, tensile modulus 20–30 GPa, fatigue resistance with max. stress up to 150 MPa to survive up to 2 million cycles | Carbon-reinforced long fiber thermoplastics | 15 | Injection molding |
| Foam | Light (density < 0.45 g/cm[*]), 55–70% compaction point, compatible with over-molding, good elastic recovery and fatigue resistance, preferably injection moldable using ICRC equipment | TPU compounds with physical blowing agents | 15 | Injection molding |
| Cosmesis | Superior mechanical properties especially in abrasion and tear resistance, excellent aging properties, injection moldable using ICRC equipment | TPU compounds with physical blowing agents | 15 | Injection molding |
| Fasteners | Easy assembly of keel elements | Stainless steel | 5 | Machining |

[&]C.I., Cost Index: This value is a percentage of the target cost (100 USD)

[*] 15% is the cost associated with assembling the prosthetic foot.

requirements are presented in Table 2. For the blade part, it appeared that carbon continuous thermoset composites, which are widely used in high-performance prosthetic feet, are necessary for the required stiffness and elastic energy restitution within the geometric thickness constraints. Thermoplastic-based materials and processes were considered to limit the cost associated with the complex shape of the ankle. A similar shape produced by traditional thermoset composite processes did not fit our target costs. Additionally thermoplastic processes such as injection molding provided full automation and low cycle times leading to cost savings. Carbon fiber reinforced Polyamide LFTs were selected due to their sufficiently high mechanical properties and reasonable cost. Injectable TPU compounds with foaming agents were selected for the foam part to preserve compatibility with the existing cosmetic shell material and achieve good adhesion.

The selected keel materials were successfully qualified by fatigue testing to ensure durability as verified with the results presented in Table 3. All the materials passed the required fatigue resistance without showing any signs of failure and thus, were selected for the final prototype manufacturing.

### 3.3 Prototype testing

For each foot, the load-deflection response was measured at different pitch angles to extract its non-linear stiffness characteristics starting from a vertical preload of 70N, as shown in Fig 8A.

**Table 3. Fatigue tests of the ankle and blade materials.**

| Part | Material | Manufacturer | Process | Stress, MPa | Failed at no. of cycles | Test stopped at no. of cycles |
|------|----------|--------------|---------|-------------|------------------------|-------------------------------|
| Ankle | Polyamide filled with more than 20% by weight Carbon Fiber LFT | EMS Chemie | Injection molding | 150 | NF[*] | 5,000,000 |
| | | | | 175 | 655,201 | - |
| | | | | 200 | 4001 | - |
| Blade | Carbon Epoxy 200 GSM Prepreg with layup (+45/-45/0$_8$)s | Gurit | Vacuum bagging and autoclave curing | 746 | NF[*] | 2,000,000 |
| | | | | 766 | 149,451 | - |
| Foam | TPU compounds with blowing agents | BASF | Injection molding | 0.9 | NF[*] | 2,000,000 |

NF[*]: Not failed under these conditions.

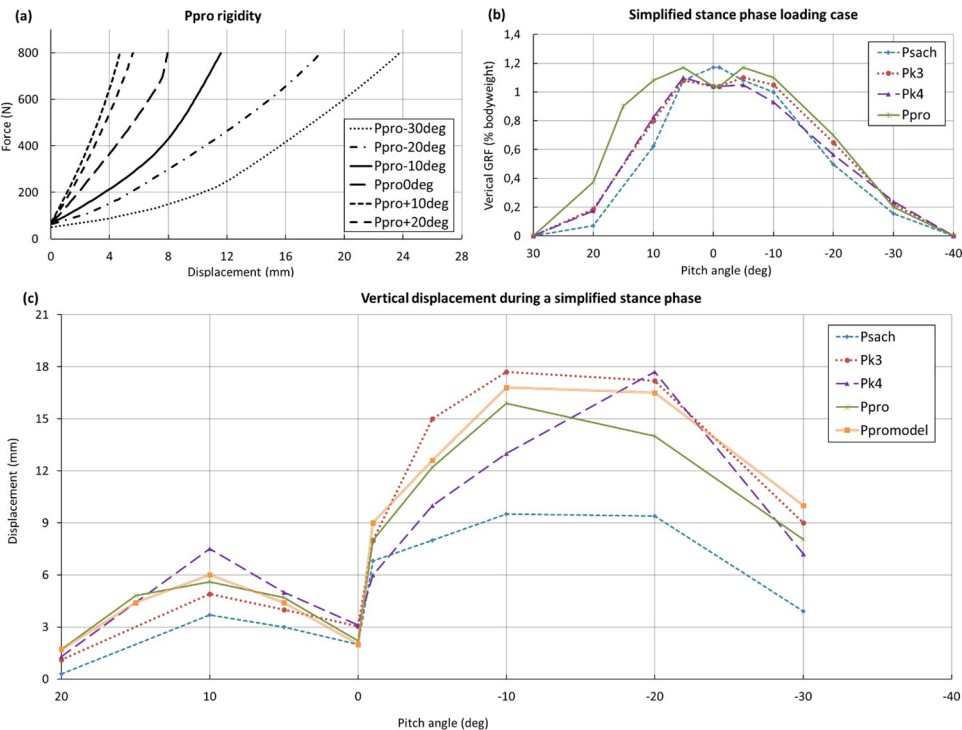

**Fig 8. Evaluation of the load-deflection response.** (a) Representation of the $P_{PRO}$ rigidity at different angles. (b) Simplified loading representation of the NvGRF. (c) Representation of the vertical deformation of different feet during a stance phase. Please note that Pk3 denotes the $P_{K3\_C}$ foot here.

The vertical deflection was recorded during mechanical testing for each pitch angle at the corresponding level of GRF measured during gait testing (Fig 8B), where 100% GRF was defined as 800N. The displacement and pitch angle chart (Fig 8C) summarizes the behavior of the feet during a complete stance phase, in other words it represents the vertical displacement observed during mechanical testing in function of the pitch angle at the corresponding force measured during the gait testing. As expected, all feet presented a higher rigidity (and lower displacements) at the heel compared to the forefoot (Fig 8A and 8C), and most of the feet showed highly non-linear stiffness characteristics (Fig 8A). $P_{PRO}$, $P_{K3\_C}$, $P_{K3\_E}$, and $P_{K4}$ exhibited higher forefoot flexibility than $P_{SACH}$ with maximal displacements in the range of 9 to 10mm (Fig 8C). Good agreement, within 1.5 mm deviation, between the FE model ($P_{promodel}$) predictions and $P_{PRO}$ (Fig 8C) was observed for the heel loading case, while the $P_{PRO}$ was slightly stiffer than expected in forefoot load cases. Overall, $P_{PRO}$ showed a moderately stiffer behavior than $P_{K4}$, its response sitting between $P_{K4}$ and $P_{K3}$, but was twice more compliant in the forefoot compared to $P_{SACH}$.

The $P_{SACH}$ measured moment (My) shows a peak at the swing phase (around 2deg), while the moment peak of the other feet appears later, around -10deg (Fig 9A) with lower values; $P_{K4}$ having the lowest one. With the measured moment and force, the center of pressure has been calculated and represented (Fig 9B). As expected, the center of pressure on the X-axis of $P_{SACH}$ shifts sooner than others at the forefoot (Fig 9B). This behavior is mainly explained by the geometry and the stiffness of the $P_{SACH}$ keel.

Tests according to ISO 10328:2016 P5 led to permanent deformations of less than 5mm at the forefoot and less than 2mm at the heel. No evidence of failure was detected. For fatigue tests, $P_{PRO}$ was loaded successfully with a sinusoidal force between 114N and 1320N for 2M

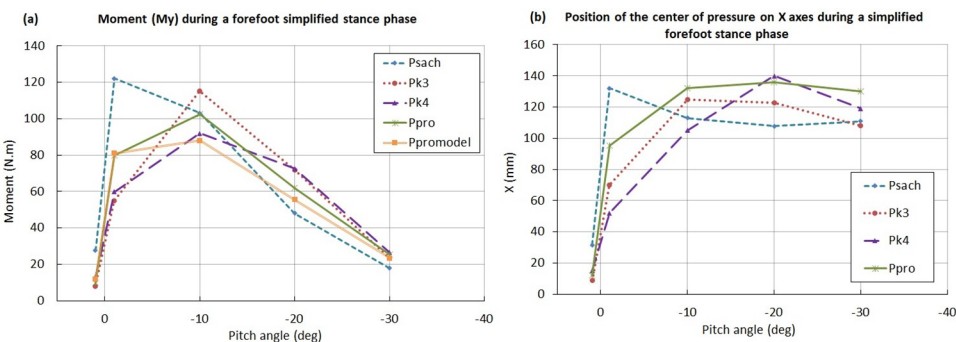

**Fig 9. Evaluation of the load-deflection response.** (a) Moment My of different during forefoot stance phase (from 0 to -30deg). (b) Representation of the position of the center of pressure on the X-axis.

cycles at the forefoot and the heel successively, without any signs of damage. The minimum test loads slightly differed from the standard recommendations because of the machine's stroke limitations. After fatigue testing, a final monotonic load test at 2240N at +15 and -20˚ was performed with no evidence of failure. The classification of the developed foot was validated by measuring the energy restitution as defined in [27].

## 3.4 Field testing

Out of the 44 days of activity monitoring, 14 days were removed from the data set because of involuntary false manipulation by the participants, sensor errors, and extreme weather (i.e., heavy rainfalls during monsoon season affected the daily activity of two participants). A total of 11142 walking bouts have been detected with 10937 bouts shorter than 30 sec, 192 bouts lasting between 30 sec and 2 min, and 13 bouts longer than 2 min. In Table 4, the results for $P_{SACH}$ and $P_{PRO}$ are compared, and the inter-trials statistics of the daily-locomotion features and the average score of the evaluation questionnaire are shown. Cadence statistics were computed over the 95% interval in order to remove outliers' errors. Finally, no statistically significant differences were observed in the distance traveled during the 6MWT and in the Q1-Q7 answers of the questionnaire (Table 4).

## 4 Discussion

The functional prototype designed and tested in this study was conceived in a relatively short time frame (1.5 years) and, although further improvements might be required, the foot displayed promising results such as mechanical performance and user acceptance. Thanks to its

**Table 4. Results from the daily-locomotion analysis and the prosthesis evaluation questionnaire[&].**

| Prosthesis | Activity monitoring | | | | | | | | | | | | Questionnaire | | | | | | |
| --- | --- | --- | --- | --- | --- | --- | --- | --- | --- | --- | --- | --- | --- | --- | --- | --- | --- | --- | --- |
| | Number of walking bouts/day | | | | Max. walking bouts duration/ day [s] | | | | Average cadence / walking bout [steps/min] | | | | Q1 | Q2 | Q3 | Q4 | Q5 | Q6 | Q7 |
| | median | min | max | N | median | min | max | N | μ | min | max | N | μ | μ | μ | μ | μ | μ | μ |
| $P_{SACH}$ | 243 | 38 | 960 | 15 | 45 | 15 | 377 | 15 | 74 | 47 | 113 | 4178 | 82 | 82 | 75 | 66 | 71 | 76 | 79 |
| $P_{PRO}$ | 414 | 149 | 795 | 15 | 87 | 25 | 695 | 15 | 73 | 45 | 111 | 5853 | 84 | 76 | 78 | 78 | 73 | 81 | 81 |
| $P_{PRO}$-$P_{SACH}$ | 171 | 111 | -165 | 0 | 42 | 10 | 318 | 0 | -1 | -2 | -2 | 1675 | 2 | -6 | 3 | 12 | 2 | 5 | 2 |

[&]The Qi indices correspond to the following questions: (1) stability while standing, (2) stability while walking, (3) powerful push-off, (4) feeling of firm contact with the ground, (5) weight of the foot, (6) comfort of the foot and (7) overall score of the prosthesis. In this table, μ represents the mean.

modular design offered by the tuning of its foam properties (i.e., density and compaction point), and the tuning of the blade thickness, the foot proposed in this study has been adapted to five different size (22, 24, 25, 26, and 28) and multiple weight classes. Moreover, the chosen materials and related processing routes allow for a mechanical performance and energy storage and release that can satisfy the K3 level requirements while reaching the targeted low manufacturing costs. Every stakeholder (i.e., ICRC and academics) shared valuable inputs and expertise to build a collective knowledge encompassing different research fields, the key findings of this interdisciplinary approach are discussed here.

We focused the output of the in-lab gait analysis on metrics directly usable in the design process (i.e., conceptualization and testing); hence the relations between the ground reaction forces and the foot orientation were extensively investigated. We identified the mechanical behavior of the $P_{K4}$ foot as the performance target based on the reported K level and the perceived comfort of the participants during the first in-lab tests. Although $P_{K4}$ is not adapted for humanitarian applications (e.g., too expensive, keel not fully encapsulated in a cosmetic shell), its mechanical efficiency in executing specific tasks became the objective targeted for the $P_{PRO}$.

The gait analysis results show significant differences ($p < .01$) in $\alpha_{pitch}$ during the loading phase for $P_{SACH}$ and $P_{K3\_C}$ compared to the targeted $P_{K4}$ foot (Table 1). In both cases, the results show that a higher vGRF was applied on the feet at positive foot strike angles; hence participants possibly felt more stable at high pitch angle (i.e., $> 10°$) with $P_{K4}$ and $P_{K3\_E}$ than with $P_{SACH}$ and $P_{K3\_C}$. Interestingly, these differences occurred even though the feet were confined within a shoe with a 10mm rear-foot cushioning, which decreased the influence of the rear-foot stiffness.

Moreover, during the pushing phase, the vGRF recorded for $P_{K4}$ decreased almost linearly with $\alpha_{pitch}$ (Fig 8A). This mechanical feature was only observed for $P_{K4}$ and could explain the feeling of "fluidity" perceived by the participants during the pushing phase. In other word this could be interpreted as a progressive feeling during the pushing phase, while a nonlinear decrease of vGRF could lead to a feeling of control lost before the end of the gait. In addition, the plantar pressure distribution confirmed that the amount of pressure on the forefoot region was higher and almost equally distributed for $P_{K4}$ and $P_{K3\_E}$ than for $P_{SACH}$ and $P_{K3\_C}$. This corroborates with the findings which infer that the mechanical properties of $P_{K4}$ and $P_{K3\_e}$ allow for a higher and more comfortable vGRF generation to drive the body forward. Although a habituation period took place after the alignment of each foot, the orthotic boots used in this study (Fig 2) indisputably increased gait variability. However, as the order of the feet was randomized, we assumed the learning bias to be similar for each foot. Also, the additional height and weight of the orthotic boots imply that higher torques were applied to the feet, which might have influenced the mechanical response of the system.

Nonetheless, based on our observations and in conformity with the K-level of the feet, the conception of $P_{PRO}$ aimed for mechanical properties bound by the observations made on the $P_{SACH}$ and $P_{K4}$ feet, acting as lower and upper-performance limits, respectively. The final in-lab gait analysis tests revealed that, during the loading phase, the participants applied significantly ($p < .01$) more vGRF at high $\alpha_{pitch}$ compared to the other feet (Fig 6) and $P_{PRO}$ exhibited a gait pattern close to $P_{K3\_E}$ in the pushing phase. Moreover, the mechanical compression tests and simulations were restricted to the angles observed during the in-lab gait analysis and to the angle set in the standards ISO10328:2016.

We combined the results from the compression tests with the vGRF observed in-lab and showed that $P_{SACH}$ (Fig 8C) was more rigid in the forefoot region compared to the other feet; a lack of vertical deformation could create instability and limit the amplitude of the pitch angle during the stance phase. From 0 to -20°, $P_{K4}$ showed the most linear vertical deformation, which corroborates with the results obtained with the participants in-lab. The shape of the keel

had a significant impact on this behavior; $P_{K4}$ has a higher and longer profile, with a longer bending length.

Compared to the high profile $P_{K4}$, the current design of $P_{PRO}$ was restricted in terms of flexibility by the 20 mm hole at the heel (insertion of the M10 bolt) and, at the forefoot, by the low-profile and split toe requirements which limit the active bending length of the blade. A careful optimization of the composite layup and thickness distribution was required to achieve both a sufficient blade flexibility while fulfilling the strength and fatigue requirements. To partially alleviate the risk of overloading the blade, we used a foam with a specific compaction point on the heel and forefoot, to bypass the load transferred through the blade in order to fulfill the overload tests required by standards. The high load observed during the heel contact phase (Fig 8B) for $P_{PRO}$ led to a moderate vertical deformation of the heel because of a higher stiffness of the heel compared to the commercial feet. This difference was not perceived as we used a cushioned heel shoe during in-lab gait tests. Indeed, shoes increase energy dissipation and reduce the incidence of injury to residual limb soft tissue [28]. To complete this rigidity analysis, we evaluated the bending moment at the ankle to reconstruct the position of the center of pressure as a function of the pitch angle during stance. It was observed that $P_{SACH}$ shows an abrupt jump of the contact point position from the heel to the toe during the rolling forward of the foot while all the other feet including $P_{PRO}$ exhibit a progressive motion of the center of pressure during stance (Fig 9B). The hysteresis during the compression test was measured to evaluate the overall energy restitution of $P_{PRO}$ (averaged on all angles and on 5 loading-unloading cycles) which was 82% when brand-new and 90% after fatigue (indicating a required run-in time of the encapsulation foam for full performance), compared to 87% for $P_{K3\_E}$ and 93% for $P_{K4}$. Finally, the eversion/inversion capacity was verified by evaluating the load required to ensure the contact on both sides of the foot at 0˚ pitch when contacting a 15˚ side slope. A closure force of less than 40% vGRF was found for $P_{PRO}$ which is comparable to that of reference prostheses such as $P_{K4}$ (approximately 300N).

The pragmatic approach using cost and performance as coupled guidelines led to find a compromise and step away from a full carbon epoxy composite design. As shown in Table 3, when mechanical properties are compared, long fiber carbon-epoxy composites are an obvious choice for a dynamic-foot but are beyond the cost target placed in our development. Instead, injection-molded long fiber thermoplastic materials used in the automotive industry were investigated and showed adequate stiffness and fatigue performance while strongly decreasing the cost for the targeted production volume. This combination of design, materials and manufacturing choices for each part in the prosthetic foot allows to reach the target performance and manufacturing cost of less than a 100 USD.

Finally, the results from the field study revealed that the participants walked more (+70%) and for more extended periods (+90%) with $P_{PRO}$ than with the $P_{SACH}$ foot, but no effect was observed on the walking cadence (Table 4). According to the evaluation questionnaire, $P_{PRO}$ received a better average evaluation for six of the seven most significant questions. Only the mean score of the perceived walking stability was lower (-6%) for the prototype foot. A potential explanation was found in the users' general comments; they claimed that the transition of body weight on the forefoot was difficult, and in some extreme cases, a slight backward torque was observed. This rigid behavior might originate from the fact that the participants were lighter and shorter than announced while planning the measurements; hence some participants were at the limit of the prosthesis's range of use. Overall, the participants were satisfied with $P_{PRO}$ (overall reported score of 81%). Although the questionnaire was inspired by the PEQ and not the actual validated PEQ, which we agree is not ideal and constitutes a limitation of this study, these are promising preliminary results, especially considering that the participants were equipped with the CR-SACH foot two weeks before the measurements, thus more

accustomed to it. Moreover, the lack of statistically significant difference in the 6MWT and questionnaire results is in accordance with previous studies comparing prostheses of different K-levels [29]. Hence, further field research and over a broader range of humanitarian environments should be carried out.

## 5 Conclusion

A multi-partner collaboration framework, including an academic program which focuses specifically on technologies and product development for low-resource settings, was established for this study. Through an interdisciplinary and functional approach, we developed a prosthetic foot that met the needs of ICRC's and the particular context of use in conflict-affected areas and LMICs.

We first characterized the mechanical behavior of several commercial feet and set our conception objectives accordingly. We showed that the pushing phase played a key role in K3-K4 feet compared to K1 ones. We then designed a prototype based on these gait analysis results, and the requirements set by the ISO 10328:2016 structural testing standard. This process required multiple design iterations to reach a functional foot design that was tested under different conditions. Moreover, we performed a rigorous cost analysis of materials and processing routes assessment to ensure low manufacturing costs for the considered production volume; hence, a careful screening of a multitude of materials was performed, including some that are generally not considered for such an application. Finally, tests carried out in the real-world showed a good performance of the prototype and paved the way for a long-term evaluation in the field.

Despite the promising results obtained some limitations must be considered, which will influence the next steps of the project. First, the number of patients who took part in the field tests: the authors consider that 11 patients is too few to really claim clinical evidence at this stage; further testing at a larger scale has to be performed to strengthen the first findings. Second, even if we mechanically tested the prototype with conditions as close as possible to the ISO standard, complete testing by an independent certified laboratory is required to ensure the developed foot complies with it.

The technical content of the project has been transferred to the ICRC, and the manufacturing processes should be industrialized in order to reach mass production and supply humanitarian actors and others active in LMICs.

## Supporting information

**S1 Dataset. Vertical GRF and pitch angles.** Holds the data obtained during the in-lab first and last round of the prosthetic feet tests. The data presents the relationship between the vertical ground reaction force (GRF) and the pitch angle for each participant and each prosthetic foot.
(XLSX)

**S2 Dataset. Cost summary and materials characterization.** This dataset presents the cost summary and results of the characterization tests obtained performed on the LFT ankle materials, composite blade materials and foam materials.
(DOCX)

**S3 Dataset. Compression tests.** This dataset presents the results used to generate Figs 8 and 9. The data summaries the measurements obtained with the compression test setup.
(XLSX)

**S4 Dataset. Field test gait analysis.** This dataset presents the results from the walking bouts detection and gait analysis perform in real-world conditions.
(XLSX)

**S1 Appendix. The questionnaire used for the in-field evaluation of the prostheses.**
(PDF)

**S1 File. Review additional info.**
(ZIP)

## Acknowledgments

Special thanks to the following people who brought expertise, supported, or directly took part in the project: Helena Texido Pedarros, Guillaume Martini, Pascal Morel, Pascal Hundt, Diko-lela Kalubi, Marc Zlot, Pierre Gauthier, Miguel Fernandes, Jean-Daniel Broillet, Thierry Bar-ras, Denis Müller, Robert Perrin, Elise Carron, Fabian Meylan, Wolfram Hang, Corentin Legris, Mr. Thanh, Mrs. Ha, all the staff at VIETCOT, patients at VIETCOT, ATMX workshop (EPFL), ATME workshop (EPFL), EMS-Chemie Holding AG, Polyone Corp., BASF SE, JR Purtec Gmbh, Gurit AG, CMP Gmbh, Fondation Philanthropia, Philip Morgan, Phalla Keo.

## Author Contributions

**Conceptualization:** Mathieu Falbriard, Grégory Huot, Mathieu Janier, Rajasundar Chandran, Michael Rechsteiner, Véronique Michaud, Joël Cugnoni, John Botsis, Klaus Schönenberger, Kamiar Aminian.

**Data curation:** Mathieu Falbriard, Grégory Huot, Mathieu Janier, Rajasundar Chandran.

**Formal analysis:** Mathieu Falbriard, Grégory Huot, Mathieu Janier, Rajasundar Chandran.

**Funding acquisition:** Grégory Huot, Véronique Michaud, Joël Cugnoni, John Botsis, Klaus Schönenberger, Kamiar Aminian.

**Investigation:** Mathieu Falbriard, Grégory Huot, Mathieu Janier, Rajasundar Chandran, Michael Rechsteiner.

**Methodology:** Mathieu Falbriard, Grégory Huot, Mathieu Janier, Rajasundar Chandran.

**Project administration:** Grégory Huot, John Botsis.

**Software:** Mathieu Falbriard, Mathieu Janier, Rajasundar Chandran.

**Supervision:** Grégory Huot, Véronique Michaud, Joël Cugnoni, John Botsis, Klaus Schönen-berger, Kamiar Aminian.

**Validation:** Mathieu Falbriard, Grégory Huot, Mathieu Janier, Rajasundar Chandran.

**Visualization:** Mathieu Falbriard, Mathieu Janier, Rajasundar Chandran.

**Writing – original draft:** Mathieu Falbriard, Mathieu Janier, Rajasundar Chandran.

**Writing – review & editing:** Grégory Huot, Michael Rechsteiner, Véronique Michaud, Joël Cugnoni, John Botsis, Klaus Schönenberger, Kamiar Aminian.

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
