## [Decision Letter · Decision Letter 0]

6 Oct 2020

PONE-D-20-28531

A user-centric approach towards the design, development, and test of an affordable dynamic prosthetic foot

PLOS ONE

Dear Dr. Falbriard,

Thank you for submitting your manuscript to PLOS ONE. After careful consideration, we feel that it has merit but does not fully meet PLOS ONE’s publication criteria as it currently stands. Therefore, we invite you to submit a revised version of the manuscript that addresses the points raised during the review process.

As included in the reviewers' comments the manuscript lacks clarity and justification as well as important details about the methodology, results and discussion of the results.

We look forward to receiving your revised manuscript.

Kind regards,

Arezoo Eshraghi, Ph.D.

Academic Editor

PLOS ONE

Journal Requirements:

2.Please provide additional details regarding participant consent. In the ethics statement in the Methods and online submission information, please ensure that you have specified (1) whether consent was informed and (2) what type you obtained (for instance, written or verbal, and if verbal, how it was documented and witnessed). If your study included minors, state whether you obtained consent from parents or guardians. If the need for consent was waived by the ethics committee, please include this information.

3. Please ensure that you refer to Figure 7 in your text as, if accepted, production will need this reference to link the reader to the figure.

4. Please note that in order to use the direct billing option the corresponding author must be affiliated with the chosen institute. Please either amend your manuscript or remove this option (via Edit Submission).

Reviewers' comments:

Reviewer's Responses to Questions

**Comments to the Author**

1. Is the manuscript technically sound, and do the data support the conclusions?

Reviewer #1: Partly

Reviewer #2: Yes

2. Has the statistical analysis been performed appropriately and rigorously? 

Reviewer #1: No

Reviewer #2: Yes

3. Have the authors made all data underlying the findings in their manuscript fully available?

Reviewer #1: Yes

Reviewer #2: Yes

4. Is the manuscript presented in an intelligible fashion and written in standard English?

Reviewer #1: Yes

Reviewer #2: Yes

5. Review Comments to the Author

Reviewer #1: This paper and work aim to address an important gap in prosthetic practice in LMICs by designing a new dynamic foot. The work involves a number of design and testing stages, as would be typically needed for the development of prosthetic componentry. The design process is in itself complex, and not all facets should be considered as research. While I can appreciate the work that has gone into developing the foot, the novelty is not entirely clear. Moreover, the paper appears to lack adequate rigour and details in many facets of the work as detailed below.

Abstract:

While it may be true that the prototype foot “showed a better performance and acceptance by users compared to a SACH foot”, did it perform adequately well to be deemed a dynamic foot as was the goal? The paper needs to clearly present this here, and in the discussion section.

Introduction:

The introduction does not adequately address the state of the science. What are the benefits of dynamic feet compared to SACH, and why is this critical for LMICs? Reasons other than affordable technology such as a lack of prosthetists should be acknowledged as a part of the limited access to prostheses in LMICs. There exist highly affordable feet such as the Niagara foot, and this background into existing solutions is not adequately acknowledged. In fact the Niagara foot is priced well below $100. Information about the manufacturing costs of feet needs to be dealt with and presented. What are the clinical and user issues with SACH feet? These limitations need to be presented.

Methods:

The authors define and use data relating to pitch angles and GRFs in the assessment and design of the prototype foot. Please provide some justification and citations to support that these metrics are in fact important aspects of dynamic foot design. Relating to this, the authors need to very clearly define and if applicable quantify what constitutes the differences between different foot classifications (K1 vs K2 vs K3). Please more clearly define what distinguishes a K3 foot from a K2 and K1 foot in terms of the functional or mechanical properties.

The FEA lacks detail about constraints, loading, meshing, convergences etc.

The optimization on line 225+ is vaguely described, and information is needed about what was actually done.

Line 292 – what is the 2% threshold based on?

Results:

The removal of 14 of 44 days is a significant amount (1/3 of data) and this could bias the results. How did the authors know that data were manipulated? What is the basis for removing data related to extreme weather? Monsoons are a normal occurrence in Vietnam and should therefore not be excluded. More so, did the monsoon days effect one foot more than the other?

Please provide some justification that bouts of walking are a good measure of prosthetic performance. The differences in bouts could be attributed to other factors. People in places such as Vietnam use walking as a primary means of mobility. They need mobility to pursue ADLs and other necessary tasks, so it seems unlikely that they could elect to walk more or less in light of these necessities, especially considering that the SACH foot appeared to work quite well for them. Please provide an explanation or hypothesis for these found differences.

Were the questionnaire results statistically significant? Please provide stats. Are the differences clinically significant?

Discussion and conclusions:

Need to explain the novelty and contributions of this work more clearly.

The cost aspect needs to be discussed in comparison to other existing foot technologies designed for LMICs

Minor:

Line 37 – remove ‘both’ since referring to 3 items

Line 73 – spelling of ‘toe’ is incorrect

Line 92 – need to specify the models and manufacturers of feet

Line 385 – what is 11’142?

Caption of Figure 6 stating the 13 and 5 participants is confusing and it is not clear how this relates to the actual sample size of 11 participants.

In table 1, the numbers do not make sense. For example, for the pitch angle at 50%, both the P3C and PK4 have the same value (22), yet they are noted as being significantly different. That is not possible. Also, the table is confusing, as it is not clear whether results are better or worse than the PK4. The direction matters and should be clear from the table or other figure.

Reviewer #2: GENERAL COMMENTS:

This paper presents a multi-disciplinary approach for designing and testing an affordable passive prosthetic foot suitable for the developing world context. It aims to create an affordable high mobility dynamic prosthetic foot to replace the currently distributed low mobility solid-ankle cushioned heel prostheses. The authors outline the design framework starting with design requirements, set through stakeholder discussions and gait studies of able-bodied walkers. Following with the development of the prosthetic foot design through material testing, structural optimization, manufacturing analysis, and mechanical testing. Lastly, they validated the designed prototype through gait studies, additional mechanical testing and a field trial.

The work described here has a lot of significance to the field in terms of the design approach as well as the resulting device. It is definitely worthy of a publication. However the work requires major revisions since additional information would be required to: support all the claims (mainly on the expected level of performance of the prosthetic foot prototype), justify the design and methodology choices (Why were this set of structural optimization target chosen, compared to standard measures such as roll-over shape or push off work? Can a foot that exhibit a certain load displacement behavior be categorized as fulfilling the WHO d4602 or K3 level performance? How close do you need to get to the P_K4 prosthetic foot to fulfill your requirements), and discussions on the gait analysis results would benefit from being put in perspective with existing literature (whole body propulsion, loading rate and peak vertical loads at heel strike, limitations of prosthetic boots usage compared to people with amputation, center of pressure progression, ankle moment etc…). In addition, the title of the work could be misleading as it refers to ‘user-centric’ approach but throughout the design process nor for the design requirements (given by the ICRC and WHO guidelines) were people with amputation consulted nor included in the development. People with amputation were included after the design of the prosthesis was completed. Lastly, minor revisions regarding the spelling, grammar, and phrasing issues would improve on the clarity of the work.

SPECIFIC COMMENTS:

Abstract:

Line 42: What specific aspects about the prosthetic foot performance was improved? By how much?

Introduction:

The authors should specify the level of performance of the current ICRC SACH foot to put in contrast with the target foot’s performance. What specifically about the current ICRC SACH foot prevents it to meet the listed requirements?

Line 54: How about the Niagara foot, which meets the cost requirement, through bolt attachement requirement as well as the K2-K3 level activity? (Ziolo and Bryant 2002, Wellens 2011).

Methods

Project Framework:

Line 94: The selected commercial prosthetic feet models should be specified to allow for reproducibility of the experiments, instead of abbreviated codes (P_K3_C).

Line 101: Which part of the in-lab biomechanical and mechanical evaluations where used to improve the model?

In-lab characterization of commercial prostheses:

Line 116: How long was the habituation period? Was all the tests for each participant with the commercial feet conducted on the same day?

Line 141: How were the shank, fore and rear foot frames defined? I assume the rear foot frame was defined using the three markers at the heel, the forefoot frames using the three marker at the forefoot and the shank frame, the four markers at the prosthetic ankle?

Line 145: Why were the pitch angle and flexion angle considered instead of the more traditional ankle angle or shank angle?

Line 151: Why were the stance phase behavior at 30%, 50% and 100% of BW specifically chosen instead of others gait events/frames?

Line 158: the x-axis of Fig 3b seems mislabeled.

Design, materials and simulation:

Line 165: was the shell included in the optimization/modelling of the prosthesis?

Line 168: How were the target performance outlined in d4602 translated into specific design requirements as the d4602 only outlines general walking aspects.

Line 170: The term ‘best gait pattern’ should be clarified. Which aspects of the gait pattern were targeted? Does this term refer as getting as close as the K4 foot? How close?

Line 182-183: the blade’s elasticity and strength were optimized but what were the optimization targets? Was the blade optimized independently of the foam and ankle part?

Line 184: How much eversion/inversion was set as the target?

Line 188: It is mentioned here that the foam density was used to accommodate different user weights, but only one foam density is later presented? Was the designed varied for user weights?

‘Different materials’ (line 193) and ‘ specimens’ (line 199) should include the details of the material that were tested/part of the selection process.

Line 220: Why was a quasi-static FE model chosen insead of a dynamic FE model that would represent the prosthetic foot loading cycle and loading rate of the ISO tests?

Line 224-228: Why was only the compression test of the selected commercial feet chosen as target for the optimization instead of the gait cycle tests? Furthermore, the gait cycle tests included walking activities such as stairs, ramps and side to side stepping. Were these represented too in the compression tests? Was the objective equally weighting the deformation, reaction force and moment ? Is the reaction force prescribed along with the plate orientation or was it an objective?

Mechanical compression tests

Should this section be presented before the design sections since these results were used as target during the design process?

Line 241:’Feet hysteresis were quantified and evaluated’ should these also appear in the results section?

Line 245: How were the different pitch angles selected?

Field Testing

Were these subjects given the same foot designed for an 80kg user as stated in the design section?

A copy of the adapted PEQ should be provided as supplemental information.

Line 292: How was the threshold of 2% selected? Is that related to a statistical analysis?

Results

Prototype design

Line 319: What was the result of the optimization, how close did the prototype design get to the target objectives?

Line 343: How were the stress level/fatigue resistance values set?

Line 363: Figure 8a caption refers to P_SACH but the graph represents P_PRO. It is unclear in Fig 8.b what test was conducted? Was the vertical displacement the same for all of the prostheses and the corresponding load was measured? The methods section only refers to loading cases being prescribed and not displacements. Was it from the biomechanical tests?

Figure 8 and 9: Why was only one P_K3 represented in these tests? Which one is it?

Discussion

Line 453: the P_PRO exhibit a more progressive motion of the center of pressure compared to the P_SACH but more abrupt compared to the other commercial feet? Does that allow the P_PRO to meet this requirement as there is a more abrupt change compared to the 3 higher level commercial feet?

Line 456: What explains the higher energy restitution after fatigue compared to the new state and how does it compare to the ther prosthetic feet? Was this test conducted with the foam and cosmetic shell?

Line 471: Does the difference between the P_SACH and P_PRO in terms of overall satisfaction significant?

Does the ankle moment, loading and displacement of the P_PRO enables the device to fulfill the k3 level requirement?

What were the limitations of the study? Did the prosthetic boot gait study match the walking pattern of prosthetic users?

6. PLOS authors have the option to publish the peer review history of their article (what does this mean?). If published, this will include your full peer review and any attached files.

Reviewer #1: No

Reviewer #2: No

---

## [Author Response · Author response to Decision Letter 0]

26 Jul 2021

Dear reviewers, thank you for your comments which we believe helped us improve our paper. We have updated the manuscript with track changes and answered your review in a different document. Note that we have also provided additional informations about the cost analysis and modelling in a separate .zip file.

Sincerely,

Mathieu Falbriard

---

## [Decision Letter · Decision Letter 1]

26 Aug 2021

PONE-D-20-28531R1

A functional approach towards the design, development, and test of an affordable dynamic prosthetic foot

PLOS ONE

Dear Dr. Falbriard,

Thank you for submitting your manuscript to PLOS ONE. After careful consideration, we feel that it has merit but does not fully meet PLOS ONE’s publication criteria as it currently stands.

I believe this is an important work that should ultimately be published. Yet as mentioned by the reviewers too, the structure of the paper just does not work well. Therefore, I would invite the authors to submit a reduced paper, or breaking up the work being presented in this manuscript in two manuscripts so that you can apply the necessary level of rigor and details.

We look forward to receiving your revised manuscript.

Kind regards,

Arezoo Eshraghi, Ph.D.

Academic Editor

PLOS ONE

Reviewers' comments:

Reviewer's Responses to Questions

**Comments to the Author**

1. If the authors have adequately addressed your comments raised in a previous round of review and you feel that this manuscript is now acceptable for publication, you may indicate that here to bypass the “Comments to the Author” section, enter your conflict of interest statement in the “Confidential to Editor” section, and submit your "Accept" recommendation.

Reviewer #1: (No Response)

Reviewer #2: (No Response)

2. Is the manuscript technically sound, and do the data support the conclusions?

Reviewer #1: Partly

Reviewer #2: Yes

3. Has the statistical analysis been performed appropriately and rigorously? 

Reviewer #1: I Don't Know

Reviewer #2: Yes

4. Have the authors made all data underlying the findings in their manuscript fully available?

Reviewer #1: Yes

Reviewer #2: Yes

5. Is the manuscript presented in an intelligible fashion and written in standard English?

Reviewer #1: Yes

Reviewer #2: Yes

6. Review Comments to the Author

Reviewer #1: First, I would like to acknowledge the importance of the work aimed at addressing a challenging global problem. Secondly, I appreciate the multifaceted nature of the approach, considering a range of aspects relevant to solving the problem. The authors present analyses related to biomechanical, structural, finite element, cost, and various aspects of the design, as well as clinical field-testing results. The main challenge with this paper, is that it tries to present all these findings, and as is evident in many of the responses from the first review, there just is not enough space to treat each of these aspects with the level of detail and rigor that (in my opinion) is needed for a research paper. As such, there continue to be too many gaps in the paper, with important information missing. Unfortunately, the manuscript reads more like a student thesis project, than a research paper that one would expect to find in PLOS.

Introduction:

“According to the World Health Organization (WHO), 80% of people with disabilities live in low- and

middle-income countries (LMICs) [1], and the number of lower-limb amputees is likely to globally increase

due to several factors such as diabetes, accidents, conflicts, and congenital disabilities [2]. Moreover, only 5

to 15 percent of people with disabilities living in LMICs have access to rehabilitation services. Prosthetic

feet (further described as "feet" for ease of reading) with advanced biomechanical features ..”

This section jumps from disability to lower-limb amputees, back to disability, and then very specific info related to feet. I think the flow of the information needs to be improved. Perhaps the information on feet deserves its own paragraph(s).

Line 58 ‘usually made of expensive materials and..’ – please be specific and provide an example of the expensive materials in question

Line 74 ‘material and production processes (e.g., 3D-printing) have shown promising’ – its not clear why 3D printing is mentioned here, since it does not appear to play a role in the project.

Line 79 – Is this manufacturing or the final cost? Most prosthetic devices have high margins, so the manufacturing cost is usually a small fraction of the final cost. In other words, many feet exist that cost under $100 to make, but are sold for many times that. Noting that it is the final cost that is important to the client. Please clarify and provide more insight about these costs.

In addition to K level, why is the weight capacity not considered along with the other design criteria

Methods:

3.2.1. – please provide information or references that speak to the validity of using this protocol (able-bodied participants using the brace) to assess foot gait biomechanics.

Line 149 – ‘The data from these units were used to empirically confirm that the previously developed algorithms for sound leg gait analysis [16,17] were functioning on the prosthetic foot 151 and could be used for field tests.’ – It’s not clear what this statement aims to communicate.

‘However, we used the median, and the interquartile range (IQR) as inter-steps statistics (i.e., calculated over all the steps regardless of the subject) as histograms and the quantile-quantile plots (Q-Q plots) suggested a non-Gaussian distribution’ – Why? Please justify.

Line 199 – ‘…minimal use of carbon fiber reinforced material to limit cost, geometrical constraints due to the’

Please elaborate on the above. In particular what is the cost of carbon fibre compared to other materials? This seems like it would be important information if one is trying to optimize costs.

Lin 216 – Here and in a number of places the authors rely fully on references that speak to a particular protocol. This is not sufficient as it requires the reader in most cases to seek information elsewhere. The manuscript should possess adequate information/details to stand alone.

Line 251 ‘The stiffness of the prototype …overloading.’ Please provide details about the manual optimization. More information is needed about how the optimization was conducted and variables applied. How is injection moulding a design constraint?

Line 303 ‘In addition, we used a foot assessment questionnaire inspired from the Prosthesis Evaluation Questionnaire (PEQ) [24] to evaluate..’ – The PEQ is a validated questionnaire and unfortunately, using questions from it is not ideal. This is a major limitation that should be clearly recognized.

Line 319 – ‘The questionnaire was composed of 14 explicit…’ – which questionnaire is this?

Results:

Line 329 – ‘This number varied between subjects as invalid steps..’ – it is typical to collect data and exclude invalid step during collection – hence extra steps can be collected to replace the mishits. Why was this not done?

Line 364 – ‘Thermoplastic-based materials and processes were considered to limit the cost

associated with the complex shape of the ankle’ Please explain what this means.

Table 2 – please explain the cost index

Table 3 – the numbers are confusing. For example, it seems that the blade failed at 149,451 cycles(?)

The data from the PEQ and 6MWT need to be presented and discussed even if the findings were not significant. The fact that the prototype foot did not perform better in the 6MWT needs to be discussed. What have other studies found with this test when comparing SACH to higher end feet?

Discussion:

Line 433 – ‘..the foot displayed promising results.’ – Such as?

Line 434 - ‘the foot proposed in this study can be adapted to a wide range of patients' sizes and weights.’ – it is unclear where in the findings this was demonstrated.

Line 454 – ‘This mechanical feature was only observed for PK4 and could explain the feeling of “fluidity” perceived by..’ – Please clarify what data/results demonstrate this perceived fluidity.

Line 484 – ‘for PPRO led to a moderate deformation because of a higher stiffness..’ deformation of what?

‘This difference was not perceived as we used a cushioned heel..’ – by what or who was it not perceived? Please refer to the relevant results/data

Line 489 – spelling of ‘two’

‘The pragmatic approach using cost and performance as coupled guidelines allowed the team to step

away from a full carbon epoxy composite design,..’ reducing the amount of carbon also seems to have reduced the performance compared to the K4 foot. So it in fact appears to be a compromise. These points should be discussed. A carbon ankle (c section) will no doubt play a role in the dynamic performance of the foot. So how much performance is lost be replacing it with an injection moulded component? It would be interesting to discuss the elastic properties in relation to typical carbon fibre ones.

‘PPRO received a better average evaluation for six of the seven most significant questions’ – please include these results in the results section.

Reviewer #2: GENERAL COMMENTS:

This paper presents a multi-disciplinary approach for designing and testing an affordable passive prosthetic foot suitable for the developing world context. It aims to create an affordable high mobility dynamic prosthetic foot to replace the currently distributed low mobility solid-ankle cushioned heel prostheses. The work described here has a lot of significance to the field in terms of the design approach as well as the resulting device. It is definitely worthy of a publication. The authors have improved and addressed some of the reviewer’s comments but minor revisions should still be completed before submission. Including additional quantifiable measures to interpret the results would improve on the rigor and clarity of the manuscript. Instead of mentioning that the model is a ‘good match’ with the experiments providing the error in displacement or moments would be beneficial. The supplementary documents should be listed/referred to throughout the manuscript when relevant instead of at the end of the manuscript only. Similarly, the load displacement hysteresis curves, questionnaire information or questionnaire results mentioned in the reviewers’ response to be present in the supplementary document were not found.

SPECIFIC COMMENTS:

Methods

Design, materials and simulation:

Line 197: the ‘best’ gait pattern is a subjective assessment, be more specific and accurate by saying for example that feet were designed to replicate the pitch angles at different instances of stance.

Materials and process selection methodology:

Refer to supplementary material 2

Mechanical simulation:

Line 261: typo, either ‘cosmesis’ or ‘cosmetic shell’

Field Testing

Refer to the supplemental information for the questionnaire and it was not found in the supporting information listed in section 8.

Results

Prototype testing

Be more specific regarding which figure was collected during gait testing and which one during the mechanical testing. From the manuscript it seems that the (a) and (c) were collected on the mechanical test setup while (b) is data from the gait testing.

Line 404: replace ‘gait GRF’ by GRF measured during gait testing to improve clarity.

Line 411: ‘good agreement’ should be quantified, model predicted the displacement within ..mm

Line 420: ‘peak at swing phase’ but the plot shown in Fig 8.a says moment during stance phase.

Figure 8a: the title during forefoot simplified stance phase could be clarify to convey the portion of stance phase where the forefoot is loaded.

I assume here too that the CoP was back calculated from the moment and loading values in the compression test?

‘the classification of the … by measuring energy restitution as defined in..” can you list the prototype’s energy restitution value compared to the other devices?

Field testing:

Were the activity monitoring and questionnaire results statistically significant? Please include this information in the result section. In addition, the supplementary data (S4) does not seem to include the questionnaire answers as stated

Discussion

Line 513: typo ‘toe’ instead of ‘tow’

How do you explain that the CoP for small pitch angles was further down the foot compared to CoP at larger pitch angles for P_SACH? It is counter-intuitive that the CoP progression would be this way in gait testing. Additional comments regarding the results would be helpful.

You mentioned that the questionnaire could only be accurately measured up to 2% in the method section (line thickness) but 3 out of the 7 questions had differences of 2% were still considered as an improvement/better answer from the participants. Should the comment regarding the improved feedback/perception only be valid for Q4 or maybe Q6 depending on the statistical analysis?

Comments to the reviewers regarding the inversion/eversion testing conducted to validate that the designed prostheses indeed enabled this functionality should be mentioned in the manuscript.

Some information about the cost model and cost breakdown included in the reviewer’s documents should be included in the discussion or results to show that indeed the $100 threshold was achieved.

7. PLOS authors have the option to publish the peer review history of their article (what does this mean?). If published, this will include your full peer review and any attached files.

Reviewer #1: No

Reviewer #2: No

---

## [Author Response · Author response to Decision Letter 1]

7 Nov 2021

We apologize that during the previous submission to the reviewers we made an error by submitting highly confidential supplementary information ( S2: material characterization) and we kindly ask to revoke those specific documents submitted previously marked confidential that includes a supplementary information word document(SI_confidential_agilis paper) and a detailed cost analysis excel sheet(Cost modelling prostheses). Due to this additional information submitted earlier we understand that the reviewers have many queries about why we have not mentioned more detailed information in our article. The only reason we could not express ourselves more freely is due to the confidential information that we need to protect to ensure the industrialization and wide outreach of this product to all the patients in vulnerable areas worldwide through the ICRC. A revised supplementary information document for the materials and cost is submitted in this version. We hope the reviewers will understand this sensitive situation and support us in this effort.

---

## [Decision Letter · Decision Letter 2]

14 Dec 2021

PONE-D-20-28531R2A functional approach towards the design, development, and test of an affordable dynamic prosthetic footPLOS ONE

Dear Dr. Falbriard,

Thank you for submitting your manuscript to PLOS ONE. After careful consideration, we feel that it has merit but does not fully meet PLOS ONE’s publication criteria as it currently stands. Therefore, we invite you to submit a revised version of the manuscript that addresses the points raised during the review process.

We look forward to receiving your revised manuscript.

Kind regards,

Arezoo Eshraghi, Ph.D.

Academic Editor

PLOS ONE

Journal Requirements:

Reviewers' comments:

Reviewer's Responses to Questions

**Comments to the Author**

1. If the authors have adequately addressed your comments raised in a previous round of review and you feel that this manuscript is now acceptable for publication, you may indicate that here to bypass the “Comments to the Author” section, enter your conflict of interest statement in the “Confidential to Editor” section, and submit your "Accept" recommendation.

Reviewer #1: (No Response)

Reviewer #2: (No Response)

2. Is the manuscript technically sound, and do the data support the conclusions?

Reviewer #1: Yes

Reviewer #2: Yes

3. Has the statistical analysis been performed appropriately and rigorously? 

Reviewer #1: I Don't Know

Reviewer #2: Yes

4. Have the authors made all data underlying the findings in their manuscript fully available?

Reviewer #1: Yes

Reviewer #2: No

5. Is the manuscript presented in an intelligible fashion and written in standard English?

Reviewer #1: Yes

Reviewer #2: Yes

6. Review Comments to the Author

Reviewer #1: -Line 30 change ‘actors’ to ‘organizations’, also would remove ‘to a growing number of people’. Or improve the flow of the sentence otherwise.

-Lines 33-35 – a reference is needed to support the assertion that there is a need for more dynamic feet. Also, it is not the low-income setting that would benefit, but rather the individuals with amputations that live in those settings. Please rephrase

-Line 35-36 – this sentence requires references, so that the reader can look up information about these other attempt

- Lines 37-46 should be better organized. Perhaps chronological order. i.e. development of design requirements, design and fabrication of foot, then testing. Also, should information should be included what population testing was done on (gait analysis, field etc)

- line 55 – should say ‘services’

-line 58 -replace ‘like’ with ‘such as’

-line 74 – replace ‘the’ with ‘the feet’. Remove extra period.

-Line 75 – ‘needs in biomechanics, durability, aspect, compatibility with the PP technology, and cost.’ This part of the sentence does not make sense. Please reword.

-Line 81 – the word ‘and’ is missing before the last point.

-Line 82 -which P level in particular is relevant or being targeted here?

-Line 85 – replace ‘prevent’ with ‘decrease’

-Line89 ‘ remove word ‘generic’ or replace with ‘main’. Also, where does the testing come in?

Line 101- rephrase to ‘To gain insight into the mechanical response of existing feet..’

Line 112 – should say’..was started’

Line 115 – please rephrase this since simulations themselves re not doing the interacting

Line 117 – iterations of what? Design interactions?

Line 162 – replace ‘over’ with ‘for’

Line 165 – Consider a more objective approach for assessing the normality of the distribution e.g. Shapiro-Wilk test. Please perform the test and provide results in the paper.

Line 168 – Should ‘time’ be plural?

Line 183 – structural integrity my be a more appropriate term than reliability.

Line 213 – a word after patient is missing. Patient characteristics?

Line 306- consider replacing the word ‘comfort’ with ‘performance’ to more accurately capture the breadth of the outcome measures used

Line 307 – what is a ‘healthy’ amputee. Please be more specific about how they were screened.

Line 312 – checking for what?

Line 313 – it’s more commonly referred to as the 6 minute walk test.

Line 315 – replace ‘inspired’ by ‘adapted’

Line 316 – ‘the feeling of the users’ is awkward wording.

Line 328 – be specific about which questionnaire is being referred to.

Line 335-36 – need to more specific about the source of the data (i.e. the gait analysis)

Line 359 – replace ‘aspect’ with ‘design’

Line 430 – how were instances when participants manipulated the data determined?

Line 432 – ‘A total of 11142 walking bouts have..’ is this for all the participants?

Line 451 – the word ‘performances’ should be singular.

Line 451 –‘ Moreover, the chosen materials and related processing routes allow for performances comparable

high-end commercial products..’ this is somewhat of a pointless statement. More useful is to state whether comparable performance was actually achieved. My personal opinion is that there is a compromise that needs to be clearly noted as the design does not perform to the same level of high-end devices.

Line 454-5 – The point of the sentence is unclear. What does the transdisciplinary nature of the work have to do with it being reported here?

Line 467 – chance ‘inside’ to ‘within’

Line 500- change to ‘during the heel…’

Line 545- ‘prototype's conception objective’ is awkward phrasing

Line 547 – change to ISO structural testing standards

Reviewer #2: GENERAL COMMENTS:

This paper presents a multi-disciplinary approach for designing and testing an affordable passive prosthetic foot suitable for the developing world context. It aims to create an affordable high mobility dynamic prosthetic foot to replace the currently distributed low mobility solid-ankle cushioned heel prostheses. The work described here has a lot of significance to the field in terms of the design approach as well as the resulting device. It is definitely worthy of a publication.

The authors have edited the manuscript with improved the clarity, details, and addressed the reviewer’s comments but minor small revisions should be implemented before submission.

SPECIFIC COMMENTS:

Introduction

Line 53: Typo ‘..’

Line 62: Please explain ‘level 3 activities’ and the remaining mobility levels (Line 74) instead of just adding the reference as they might be confused with the Medicare K levels introduced later.

Line 64: ‘ Due to its high profile’, what is considered a high profile and what was the prosthetic height target set for the author’s prosthetic feet?

Methods

Line 223: Typo ‘ this research, Scalability’

Results

Table 3: Typo ‘ Carbon Fibre’

Discussion

Line 513: “A closure force of less than 40% vGRF was found for PPRO which is comparable

513 to that of reference prostheses such as PK4.” What was the closure force of PK4 so that the reader can also deem it comparable?

Questionnaire supplemental material: In the questionnaire questions, 14 items are listed but only 7 are reported in the results both in the supplemental material as well as in the manuscript. Why were the other 7 questions omitted?

The hysteresis curves have should been included in supplemental material as they are mentioned in the test methodology and discussions. In addition, are the energy restitution averaged across the different loading cases or only taken from a single loading case?

7. PLOS authors have the option to publish the peer review history of their article (what does this mean?). If published, this will include your full peer review and any attached files.

Reviewer #1: No

Reviewer #2: No

---

## [Author Response · Author response to Decision Letter 2]

23 Mar 2022

Dear reviewers, thank you for taking the time to comment on our study. We believe your feedback and questions helped improve the manuscript. Please find the detailed answers in the "Answer to reviewers" document attached to this submission.

Sincerely

Mathieu Falbriard, Ph.D.

---

## [Editor Report · Decision Letter 3]

25 Mar 2022

A functional approach towards the design, development, and test of an affordable dynamic prosthetic foot

PONE-D-20-28531R3

Dear Dr. Falbriard,

We’re pleased to inform you that your manuscript has been judged scientifically suitable for publication and will be formally accepted for publication once it meets all outstanding technical requirements.

Kind regards,

Arezoo Eshraghi, Ph.D.

Academic Editor

PLOS ONE
---

## [Editor Report · Acceptance letter]

29 Apr 2022

PONE-D-20-28531R3 

A functional approach towards the design, development, and test of an affordable dynamic prosthetic foot 

Dear Dr. Falbriard:

I'm pleased to inform you that your manuscript has been deemed suitable for publication in PLOS ONE. Congratulations! Your manuscript is now with our production department. 

Kind regards, 

on behalf of

Dr. Arezoo Eshraghi 

Academic Editor

PLOS ONE